# Controlling Neural Level Sets

**Matan Atzmon, Niv Haim, Lior Yariv, Ofer Israelov, Haggai Maron, Yaron Lipman**
Weizmann Institute of Science
Rehovot, Israel

## Abstract

The level sets of neural networks represent fundamental properties such as decision boundaries of classifiers and are used to model non-linear manifold data such as curves and surfaces. Thus, methods for controlling the neural level sets could find many applications in machine learning.

In this paper we present a simple and scalable approach to directly control level sets of a deep neural network. Our method consists of two parts: (i) sampling of the neural level sets, and (ii) relating the samples' positions to the network parameters. The latter is achieved by a *sample network* that is constructed by adding a single fixed linear layer to the original network. In turn, the sample network can be used to incorporate the level set samples into a loss function of interest.

We have tested our method on three different learning tasks: improving generalization to unseen data, training networks robust to adversarial attacks, and curve and surface reconstruction from point clouds. For surface reconstruction, we produce high fidelity surfaces directly from raw 3D point clouds. When training small to medium networks to be robust to adversarial attacks we obtain robust accuracy comparable to state-of-the-art methods.

## 1 Introduction

The level sets of a Deep Neural Network (DNN) are known to capture important characteristics and properties of the network. A popular example is when the network $F(x; \theta) : \mathbb{R}^d \times \mathbb{R}^m \to \mathbb{R}^l$ represents a classifier, $\theta$ are its learnable parameters, $f_i(x; \theta)$ are its logits (the outputs of the final linear layer), and the level set

$$\mathcal{S}(\theta) = \left\{ x \in \mathbb{R}^d \ \middle| \ f_j - \max_{i \neq j} \{f_i\} = 0 \right\} \tag{1}$$

represents the decision boundary of the $j$-th class. Another recent example is modeling a manifold (e.g., a curve or a surface in $\mathbb{R}^3$) using a level set of a neural network (e.g., [24]). That is,

$$\mathcal{S}(\theta) = \left\{ x \in \mathbb{R}^d \ \middle| \ F = 0 \right\} \tag{2}$$

represents (generically) a manifold of dimension $d - l$ in $\mathbb{R}^d$.

The goal of this work is to provide practical means to directly control and manipulate *neural level sets* $\mathcal{S}(\theta)$, as exemplified in Equations 1, 2. The main challenge is how to incorporate $\mathcal{S}(\theta)$ in a differentiable loss function. Our key observation is that given a sample $p \in \mathcal{S}(\theta)$, its position can be associated to the network parameters: $p = p(\theta)$, in a *differentiable* and *scalable* way. In fact, $p(\theta)$ is itself a neural network that is obtained by an addition of a *single* linear layer to $F(x; \theta)$; we call these networks *sample networks*. Sample networks, together with an efficient mechanism for sampling the level set, $\{p_j(\theta)\} \subset \mathcal{S}(\theta)$, can be incorporated in general loss functions as a proxy for the level set $\mathcal{S}(\theta)$.

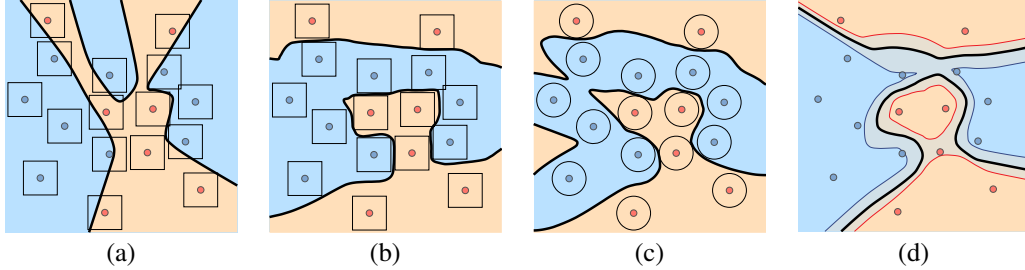

<div align="center">(a)        (b)        (c)        (d)</div>

Figure 1: Our method applied to binary classification in 2D. Blue and red dots represent positive and negative examples respectively. (a) standard cross-entropy loss baseline; (b) using our method to move the decision boundary at least $\varepsilon$ away from the training set in $L_\infty$ norm; (c) same for $L_2$ norm; (d) a geometrical adaptation of SVM soft-margin loss to neural networks using our method, the $+1, -1$ level sets are marked in light red and blue, respectively. Note that in (b),(c),(d) we achieve decision boundaries that seem to better explain the training examples compared to (a).

We apply our method of controlling the neural level sets to two seemingly different learning tasks: controlling decision boundaries of classifiers (Equation 1) and reconstructing curves and surfaces from point cloud data (Equation 2).

At first, we use our method to improve the generalization and adversarial robustness in classification tasks. In these tasks, the distance between the training examples and the decision boundary is an important quantity called the *margin*. Margins are traditionally desired to be as large as possible to improve generalization [7, 10] and adversarial robustness [10]. Usually, margins are only controlled indirectly by loss functions that measure network output values of training examples (e.g., cross entropy loss). Recently, researchers have been working on optimizations with more direct control of the margin using linearization techniques [13, 21, 10], regularization [26], output-layer margin [27], or using margin gradients [9]. We suggest controlling the margin by sampling the decision boundary, constructing the sample network, and measuring distances between samples and training examples directly. By applying this technique to train medium to small-size networks against adversarial perturbations we achieved comparable robust accuracy to state-of-the-art methods.

To improve generalization when learning from small amounts of data, we devise a novel geometrical formulation of the soft margin SVM loss to neural networks. This loss aims at directly increasing the *input space* margin, in contrast to standard deep network hinge losses that deal with output space margin [28, 27]. The authors of [10] also suggest to increase input space margin to improve generalization. Figure 1 shows 2D examples of training our adversarial robustness and geometric SVM losses for networks.

In a different application, we use our method for the reconstruction of manifolds such as curves and surfaces in $\mathbb{R}^3$ from point cloud data. The usage of neural networks for the modeling of surfaces has recently become popular [12, 29, 6, 2, 24, 22]. There are two main approaches: parametric and implicit. The parametric approach uses networks as parameterization functions to the manifolds [12, 29]. The implicit approach represents the manifold as a level set of a neural network [24, 6, 22]. So far, implicit representations were learned using regression, given a signed distance function or occupancy function computed directly from a ground truth surface. Unfortunately, for raw point clouds in $\mathbb{R}^3$, computing the signed distance function or an occupancy function is a notoriously difficult task [4]. In this paper we show that by using our sample network to control the neural level sets we can reconstruct curves and surfaces directly from point clouds in $\mathbb{R}^3$.

Lastly, to theoretically justify neural level sets for modeling manifolds or arbitrary decision boundaries, we prove a geometric version of the universality property of MLPs [8, 14]: any piecewise linear hyper-surface in $\mathbb{R}^d$ (i.e., a $d-1$ manifold built out of a finite number of linear pieces, not necessarily bounded) can be precisely represented as a neural level set of a suitable MLP.

## 2 Sample Network

Given a neural network $F(x; \theta) : \mathbb{R}^d \times \mathbb{R}^m \to \mathbb{R}^l$ its $0 \in \mathbb{R}^l$ level set is defined by

$$\mathcal{S}(\theta) := \left\{ x \ \middle| \ F(x; \theta) = 0 \right\}. \tag{3}$$

We denote by $D_x F(p; \theta) \in \mathbb{R}^{l \times d}$ the matrix of partial derivatives of $F$ with respect to $x$. Assuming that $\theta$ is fixed, $F(p; \theta) = 0$ and that $D_x F(p; \theta)$ is of full rank $l$ ($l \ll d$), a corollary of the Implicit Function Theorem [15] implies that $\mathcal{S}(\theta)$ is a $d - l$ dimensional manifold in the vicinity of $p \in \mathcal{S}(\theta)$.

Our goal is to incorporate the neural level set $\mathcal{S}(\theta)$ in a differentiable loss. We accomplish that by performing the following procedure at each training iteration: (i) Sample $n$ points on the level set: $p_i \in \mathcal{S}(\theta)$, $i \in [n]$; (ii) Build the sample network $p_i(\theta)$, $i \in [n]$, by adding a fixed linear layer to the network $F(x; \theta)$; and (iii) Incorporate the sample network in a loss function as a proxy for $\mathcal{S}(\theta)$.

### 2.1 Level set sampling

To sample $\mathcal{S}(\theta)$ at some $\theta = \theta_0$, we start with a set of $n$ points $p_i$, $i \in [n]$ sampled from some probability measure in $\mathbb{R}^d$. Next, we perform generalized Newton iterations [3] to move each point $p$ towards $\mathcal{S}(\theta_0)$:

$$p^{\text{next}} = p - D_x F(p; \theta_0)^\dagger F(p; \theta_0), \tag{4}$$

where $D_x F(p, \theta_0)^\dagger$ is the Moore-Penrose pseudo-inverse of $D_x F(p, \theta_0)$. The generalized Newton step solves the under-determined ($l \ll d$) linear system $F(p; \theta_0) + D_x F(p; \theta_0)(p^{\text{next}} - p) = 0$. To motivate this particular solution choice we show that the generalized Newton step applied to a linear function is an orthogonal projection onto its zero level set (see proof in the supplementary material):

**Lemma 1.** *Let $\ell(x) = Ax + b$, $A \in \mathbb{R}^{l \times d}$, $b \in \mathbb{R}^l$, $\ell < d$, and $A$ is of full rank $l$. Then Equation 4 applied to $F(x) = \ell(x)$ is an orthogonal projection on the zero level set of $\ell$, namely, on $\{x \mid \ell(x) = 0\}$.*

For $l > 1$ the computation of $D_x F(p; \theta_0)^\dagger$ requires inverting an $l \times l$ matrix; in this paper $l \in \{1, 2\}$. The directions $D_x F(p_i; \theta_0)$ can be computed efficiently using back-propagation where the points $p_i$, $i \in [n]$ are grouped into batches. We performed $10 - 20$ iterations of Equation 4 for each $p_i$, $i \in [n]$.

**Scalar networks.**  For scalar networks, i.e., $l = 1$, $D_x F \in \mathbb{R}^{1 \times d}$, a direct computation shows that

$$D_x F(p; \theta_0)^\dagger = \frac{D_x F(p; \theta_0)^T}{\|D_x F(p; \theta_0)\|^2}. \tag{5}$$

That is, the point $p$ moves towards the level set $\mathcal{S}(\theta_0)$ by going in the direction of the steepest descent (or ascent) of $F$.

It is worth mentioning that the projection-on-level-sets formula in the case of $l = 1$ has already been developed in [23] and was used to find adversarial perturbations; our result generalizes to the intersection of several level sets and shows that this procedure is an instantiation of the generalized Newton algorithm.

The generalized Newton method (similarly to Newton method) is not guaranteed to find a point on the zero level set. We denote by $c_i = F(p_i; \theta_0)$ the level set values of the final point $p_i$; in the following, we use also points that failed to be projected with their level set $c_i$. Furthermore, we found that the generalized Newton method usually does find zeros of neural networks but these can be far from the initial projection point. Other, less efficient but sometimes more robust ways to project on neural level sets could be using gradient descent on $\|F(x; \theta)\|$ or zero finding in the direction of a successful PGD attack [17, 20] as done in [9]. In our robust networks application (Section 3.2) we have used a similar approach with the false position method.

**Relation to Elsayed et al. [10].**  The authors of [10] replace the margin distance with distance to the linearized network, while our approach is to directly sample the actual network's level set and move it explicitly. Specifically, for $L_2$ norm ($p = 2$) Elsayed's method is similar to our method using a single generalized Newton iteration, Equation 4.

## 2.2 Differentiable sample position

Next, we would like to relate each sample $p$, belonging to some level set $F(p; \theta_0) = c$, to the network parameters $\theta$. Namely, $p = p(\theta)$. The functional dependence of a sample $p$ on $\theta$ is defined by $p(\theta_0) = p$ and $F(p(\theta); \theta) = c$, for $\theta$ in some neighborhood of $\theta_0$. The latter condition ensures that $p$ stays on the $c$ level set as the network parameters $\theta$ change. As only first derivatives are used in the optimization of neural networks, it is enough to replace this condition with its first-order version. We get the following two equations:

$$p(\theta_0) = p \quad ; \quad \frac{\partial}{\partial \theta}\Big|_{\theta=\theta_0} F(p(\theta); \theta) = 0. \tag{6}$$

Using the chain rule, the second condition in Equation 6 reads:

$$D_x F(p, \theta_0) D_\theta p(\theta_0) + D_\theta F(p, \theta_0) = 0. \tag{7}$$

This is a system of linear equations with $d \times m$ unknowns (the components of $D_\theta p(\theta_0)$) and $l \times m$ equations. When $d > l$, this linear system is under-determined. Similarly to what is used in the generalized Newton method, a minimal norm solution is given by the Moore-Penrose inverse:

$$D_\theta p(\theta_0) = -D_x F(p, \theta_0)^\dagger D_\theta F(p, \theta_0). \tag{8}$$

The columns of the matrix $D_\theta p(\theta_0) \in \mathbb{R}^{d \times m}$ describe the velocity of $p(\theta)$ w.r.t. each of the parameters in $\theta$. The pseudo-inverse selects the minimal norm solution that can be shown to represent, in this case, a movement in the orthogonal direction to the level set (see supplementary material for a proof). We reiterate that for scalar networks, where $l = 1$, $D_x F(p_i; \theta_0)^\dagger$ has a simple closed-form expression, as shown in Equation 5.

**The sample network.** A possible simple solution to Equation 6 would be to use the linear function $p(\theta) = p + D_\theta p(\theta_0)(\theta - \theta_0)$, with $D_\theta p(\theta_0)$ as defined in Equation 8. Unfortunately, this would require storing $D_\theta p(\theta_0)$, using at-least $\mathcal{O}(m)$ space (i.e., the number of network parameters), for every projection point $p$. (We assume the number of output channels $l$ of $F$ is constant, which is the case in this paper.) A much more efficient solution is

$$p(\theta) = p - D_x F(p; \theta_0)^\dagger [F(p; \theta) - c], \tag{9}$$

that requires storing $D_x F(p; \theta_0)^\dagger$, using only $\mathcal{O}(d)$ space, where $d$ is the input space dimension, for every projection point $p$. Furthermore, Equation 9 allows an efficient implementation with a single network

$$G(p, D_x F(p; \theta_0)^\dagger; \theta) := p(\theta).$$

We call $G$ the *sample network*. Note that a collection of samples $p_i$, $i \in [n]$ can be treated as a batch input to $G$.

## 3 Incorporation of samples in loss functions

Once we have the sample network $p_i(\theta)$, $i \in [n]$, we can incorporate it in a loss function to control the neural level set $\mathcal{S}(\theta)$ in a desired way. We give three examples in this paper.

### 3.1 Geometric SVM

Support-vector machine (SVM) is a model which aims to train a linear binary classifier that would generalize well to new data by combining the hinge loss and a large margin term. It can be interpreted as encouraging large distances between training examples and the decision boundary. Specifically, the soft SVM loss takes the form [7]:

$$\text{loss}(w, b) = \frac{1}{N} \sum_{j=1}^{N} \max\{0, 1 - y_j \ell(x_j; w, b)\} + \lambda \|w\|_2^2, \tag{10}$$

where $(x_j, y_j) \in \mathbb{R}^d \times \{-1, 1\}$, $j \in [N]$ is the binary classification training data, and $\ell(x; w, b) = w^T x + b$, $w \in \mathbb{R}^d$, $b \in \mathbb{R}$ is the linear classifier. We would like to generalize Equation 10 to a deep network $F : \mathbb{R}^d \times \mathbb{R}^m \to \mathbb{R}$ towards the goal of increasing the network's *input space margin*, which is defined as the minimal distance of training examples to the decision boundary $\mathcal{S}(\theta) = \{x \in \mathbb{R}^d \mid F(x; \theta) = 0\}$. Note that this is in strong contrast to standard deep network hinge

loss that works with the *output space margin* [28, 27], namely, measuring differences of output logits when evaluating the network on training examples. For that reason, this type of loss function does not penalize small input space margin, so long as it doesn't damage the output-level classification performance on the training data. Using the input margin over the output margin may also provide robustness to perturbations [10].

We now describe a new, geometric formulation of Equation 10, and use it to define the soft-SVM loss for general neural networks. In the linear case, the following quantity serves as the margin:

$$\|w\|_2^{-1} = \mathrm{d}(\mathcal{S}(\theta), \mathcal{S}_1(\theta)) = \mathrm{d}(\mathcal{S}(\theta), \mathcal{S}_{-1}(\theta))$$

where $\mathcal{S}_t(\theta) = \left\{x \in \mathbb{R}^d \mid F(x; \theta) = t\right\}$, and $\mathrm{d}(\mathcal{S}(\theta), \mathcal{S}_t(\theta))$ is the distance between the level sets, which are two parallel hyper-planes. In the general case, however, level sets are arbitrary hyper-surfaces which are not necessarily equidistant (i.e., the distance when traveling from $\mathcal{S}$ to $\mathcal{S}_t$ does not have to be constant across $\mathcal{S}$). Hence, for each data sample $x$, we define the following margin function:

$$\Delta(x; \theta) = \min \left\{ \mathrm{d}\big(p(\theta), \mathcal{S}_1(\theta)\big), \mathrm{d}\big(p(\theta), \mathcal{S}_{-1}(\theta)\big) \right\},$$

where $p(\theta)$ is the sample network of the projection of $x$ onto $\mathcal{S}(\theta_0)$. Additionally, note that in the linear case: $\left|w^T x + b\right| = \mathrm{d}(x, \mathcal{S}(\theta))/\Delta(x; \theta)$. With these definitions in mind, Equation 10 can be given the geometric generalized form:

$$\mathrm{loss}(w, b) = \frac{1}{N} \sum_{j=1}^{N} \max \left\{ 0, 1 - \mathrm{sign}(y_j F(x_j; \theta)) \frac{\mathrm{d}(x_j, p_j)}{\Delta(x_j; \theta)} \right\} + \frac{\lambda}{N} \sum_{j=1}^{N} \Delta(x_j; \theta)^\alpha, \quad (11)$$

where $F(x; \theta)$ is a general classifier (such as a neural network, in our applications). Note that in the case where $F(x; \theta)$ is affine, $\alpha = -2$ and $\mathrm{d} = L_2$, Equation 11 reduces back to the regular SVM loss, Equation 10. Figure 1d depicts the result of optimizing this loss in a 2D case, i.e., $x_j \in \mathbb{R}^2$; the light blue and red curves represent $\mathcal{S}_{-1}$ and $\mathcal{S}_1$.

## 3.2 Robustness to adversarial perturbations

The goal of robust training is to prevent a change in a model's classification result when small perturbations are applied to the input. Following [20] the attack model is specified by some set $S \subset \mathbb{R}^d$ of allowed perturbations; in this paper we focus on the popular choice of $L_\infty$ perturbations, that is $S$ is taken to be the $\varepsilon$-radius $L_\infty$ ball, $\{x \mid \|x\|_\infty \leq \varepsilon\}$. Let $(x_j, y_j) \in \mathbb{R}^d \times \mathcal{L}$ denote training examples and labels, and let $\mathcal{S}^j(\theta) = \left\{x \in \mathbb{R}^d \mid F^j(x; \theta) = 0\right\}$, where $F^j(x; \theta) = f_j - \max_{i \neq j} f_i$, the decision boundary of label $j$. We define the loss

$$\mathrm{loss}(\theta) = \frac{1}{N} \sum_{j=1}^{N} \lambda_j \max \left\{ 0, \varepsilon_j - \mathrm{sign}(F^{y_j}(x_j; \theta)) \mathrm{d}(x_j, \mathcal{S}^{y_j}(\theta)) \right\}, \quad (12)$$

where $\mathrm{d}(x, \mathcal{S}^j)$ is some notion of a distance between $x$ and $\mathcal{S}^j$, e.g., $\min_{y \in \mathcal{S}^j} \|x - y\|_p$ or $\int_{\mathcal{S}^j(\theta)} \|x - y\|_p \, d\mu(y)$, $d\mu$ is some probability measure on $\mathcal{S}^j(\theta)$. The parameter $\lambda_j$ controls the weighting between correct (i.e., $F^{y_j}(x_j; \theta) > 0$) and incorrect (i.e., $F^{y_j}(x_j; \theta) < 0$) classified samples. We fix $\lambda_j = 1$ for incorrectly classified samples and set $\lambda_j$ to be the same for correctly classified samples; The parameter $\varepsilon_j$ controls the desired target distances; Similarly to [10], the idea of this loss is: (i) if $x_j$ is incorrectly classified, pull the decision boundary $\mathcal{S}^{y_j}$ toward $x_j$; (ii) if $x_j$ is classified correctly, push the decision boundary $\mathcal{S}^{y_j}$ to be within a distance of at-least $\varepsilon_j$ from $x_j$.

In our implementation we have used $\mathrm{d}(x, \mathcal{S}^j) = \rho(x, p(\theta))$, where $p = p(\theta_0) \in \mathcal{S}^j$ is a sample of this level set; $\rho(x, p) = \left|x^{i_*} - p^{i_*}\right|$, and $x^{i_*}, p^{i_*}$ denote the $i_*$-th coordinates of $x, p$ (resp.), $i_* = \arg\max_{i \in [d]} |D_x F(p; \theta_0)|$. This loss encourages $p(\theta)$ to move in the direction of the axis (i.e., $i_*$) that corresponds to the largest component in the 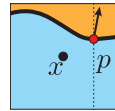 gradient $D_x F(p; \theta_0)$. Intuitively, as $\rho(x, p) \leq \|x - p\|_\infty$, the loss pushes the level set $\mathcal{S}^j$ to leave the $\varepsilon_j$-radius $L_\infty$-ball in the direction of the axis that corresponds to the maximal speed of $p(\theta)$. The inset depicts an example where this distance measure is more effective than the standard $L_\infty$ distance: $D_x F(p; \theta)$ is shown as black arrow and the selected axis $i_*$ as dashed line.

In the case where the distance function $\min_{y \in \mathcal{S}(\theta)} \|x - y\|_p$ is used, the computation of its gradient using Equation 8 coincides with the gradient derivation of [9] up to a sign difference. Still, our derivation allows working with general level set points (i.e., not just the closest) on the decision boundary $\mathcal{S}^j$, and our sample network offers an efficient implementation of these samples in a loss function. Furthermore, we use the same loss for both correctly and incorrectly classified examples.

### 3.3 Manifold reconstruction

**Surface reconstruction.** Given a point cloud $\mathcal{X} = \{x_j\}_{j=1}^N \subset \mathbb{R}^d$ that samples, possibly with noise, some surface $\mathcal{M} \subset \mathbb{R}^3$, our goal is to find parameters $\theta$ of a network $F : \mathbb{R}^3 \times \mathbb{R}^m \to \mathbb{R}$, so that the neural level set $\mathcal{S}(\theta)$ approximates $\mathcal{M}$. Even more desirable is to have $F$ approximate the signed distance function to the unknown surface sampled by $\mathcal{X}$. To that end, we would like the neural level set $\mathcal{S}_t(\theta)$, $t \in \mathcal{T}$ to be of distance $|t|$ to $\mathcal{X}$, where $\mathcal{T} \subset \mathbb{R}$ is some collection of desired level set values. Let $\mathrm{d}(x, \mathcal{X}) = \min_{j \in [N]} \|x - x_j\|_2$ be the distance between $x$ and $\mathcal{X}$. We consider the reconstruction loss

$$\mathrm{loss}(\theta) = \sum_{t \in \mathcal{T}} \left[ \int_{\mathcal{S}_t(\theta)} \left| \mathrm{d}(x, \mathcal{X}) - |t| \right|^p dv(x) \right]^{\frac{1}{p}} + \frac{\lambda}{N} \sum_{j=1}^N |F(x_j; \theta)|, \tag{13}$$

where $dv(x)$ is the normalized volume element on $\mathcal{S}_t(\theta)$ and $\lambda > 0$ is a parameter. The first part of the loss encourages the $t$ level set of $F$ to be of distance $|t|$ to $\mathcal{X}$; note that for $t = 0$ this reconstruction error was used in level set surface reconstruction methods [33]. The second part of the loss penalizes samples $\mathcal{X}$ outside the zero level set $\mathcal{S}(\theta)$.

**Curve reconstruction.** In case of approximating a manifold $\mathcal{M} \subset \mathbb{R}^d$ with co-dimension greater than 1, e.g., a curve in $\mathbb{R}^3$, one cannot expect $F$ to approximate the signed distance function as no such function exists. Instead, we model the manifold via the level set of a vector-valued network $F : \mathbb{R}^d \times \mathbb{R}^m \to \mathbb{R}^l$ whose zero level set is an intersection of $l$ hyper-surfaces. As explained in Section 2, this generically defines a $d - l$ manifold. In that case we used the loss in Equation 13 with $\mathcal{T} = \{0\}$, namely, only encouraging the zero level set to be as close as possible to the samples $\mathcal{X}$.

## 4 Universality

To theoretically support the usage of neural level sets for modeling manifolds or controlling decision boundaries we provide a geometric universality result for multilayer perceptrons (MLP) with ReLU activations. That is, the level sets of MLPs can represent any watertight piecewise linear hyper-surface (i.e., manifolds of co-dimension 1 in $\mathbb{R}^d$ that are boundaries of $d$-dimensional polytopes). More specifically, we prove:

**Theorem 1.** *Any watertight, not necessarily bounded, piecewise linear hypersurface $\mathcal{M} \subset \mathbb{R}^d$ can be exactly represented as the neural level set $\mathcal{S}$ of a multilayer perceptron with ReLU activations, $F : \mathbb{R}^d \to \mathbb{R}$.*

The proof of this theorem is given in the supplementary material. Note that this theorem is a geometrical version of Theorem 2.1 in [1], asserting that MLPs with ReLU activations can represent any piecewise linear continuous function.

## 5 Experiments

### 5.1 Classification generalization

In this experiment, we show that when training on small amounts of data, our geometric SVM loss (see Equation 11) generalizes better than the cross entropy loss and the hinge loss. Experiments were done on three datasets: MNIST [18], Fashion-MNIST [31] and CIFAR10 [16]. For all datasets we arbitrarily merged the labels into two classes, resulting in a binary classification problem. We randomly sampled a fraction of the original training examples and evaluated on the original test set.

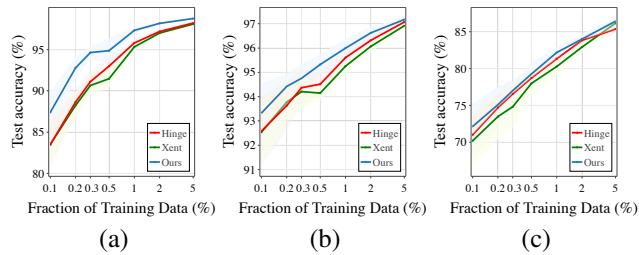

Figure 2: Generalization from small fractions of the data.

| Method | Dataset | Arch. | Attack | $\varepsilon_{\text{train}}$ | Test Acc. | Rob. Acc. Xent | Rob. Acc. Margin |
|---|---|---|---|---|---|---|---|
| Standard | MNIST | ConvNet-4a | PGD$^{40}$($\varepsilon_{\text{attack}} = 0.3$) | - | 99.34% | 13.59% | 0.00% |
| Madry et al. [20] | MNIST | ConvNet-4a | PGD$^{40}$($\varepsilon_{\text{attack}} = 0.3$) | 0.3 | 99.35% | 96.04% | 96.11% |
| Madry et al. [20] | MNIST | ConvNet-4a | PGD$^{40}$($\varepsilon_{\text{attack}} = 0.3$) | 0.4 | 99.16% | 96.54% | 96.53% |
| TRADES [32] | MNIST | ConvNet-4a | PGD$^{40}$($\varepsilon_{\text{attack}} = 0.3$) | 0.3 | 98.97% | 96.75% | 96.74% |
| TRADES [32] | MNIST | ConvNet-4a | PGD$^{40}$($\varepsilon_{\text{attack}} = 0.3$) | 0.4 | 98.62% | 96.78% | 96.76% |
| Ours | MNIST | ConvNet-4a | PGD$^{40}$($\varepsilon_{\text{attack}} = 0.3$) | 0.4 | 99.35% | 99.23% | 97.35% |
| Standard | CIFAR10 | ConvNet-4b | PGD$^{20}$ ($\varepsilon_{\text{attack}} = 0.031$) | - | 83.67% | 0.00% | 0.00% |
| Madry et al. [20] | CIFAR10 | ConvNet-4b | PGD$^{20}$ ($\varepsilon_{\text{attack}} = 0.031$) | 0.031 | 71.86% | 39.84% | 38.18% |
| Madry et al. [20] | CIFAR10 | ConvNet-4b | PGD$^{20}$ ($\varepsilon_{\text{attack}} = 0.031$) | 0.045 | 63.66% | 41.53% | 39.13% |
| TRADES [32] | CIFAR10 | ConvNet-4b | PGD$^{20}$ ($\varepsilon_{\text{attack}} = 0.031$) | 0.031 | 71.24% | 41.89% | 38.4% |
| TRADES [32] | CIFAR10 | ConvNet-4b | PGD$^{20}$ ($\varepsilon_{\text{attack}} = 0.031$) | 0.045 | 68.24% | 42.04% | 38.18% |
| Ours | CIFAR10 | ConvNet-4b | PGD$^{20}$ ($\varepsilon_{\text{attack}} = 0.031$) | 0.045 | 71.96% | 38.45% | 38.54% |
| Standard | CIFAR10 | ResNet-18 | PGD$^{20}$ ($\varepsilon_{\text{attack}} = 0.031$) | - | 93.18% | 0.00% | 0.00% |
| Madry et al. [20] | CIFAR10 | ResNet-18 | PGD$^{20}$ ($\varepsilon_{\text{attack}} = 0.031$) | 0.031 | 81.0% | 47.29% | 46.58% |
| Madry et al. [20] | CIFAR10 | ResNet-18 | PGD$^{20}$ ($\varepsilon_{\text{attack}} = 0.031$) | 0.045 | 74.97% | 49.84% | 48.02% |
| TRADES [32] | CIFAR10 | ResNet-18 | PGD$^{20}$ ($\varepsilon_{\text{attack}} = 0.031$) | 0.031 | 83.04% | 53.31% | 51.36% |
| TRADES [32] | CIFAR10 | ResNet-18 | PGD$^{20}$ ($\varepsilon_{\text{attack}} = 0.031$) | 0.045 | 79.52% | 53.49% | 51.22% |
| Ours | CIFAR10 | ResNet-18 | PGD$^{20}$ ($\varepsilon_{\text{attack}} = 0.031$) | 0.045 | 81.3% | 79.74% | 43.17% |

Table 1: Results of different $L_\infty$-bounded attacks on models trained using our method (described in Section 3.2) compared to other methods.

Due to the variability in the results, we rerun the experiment 100 times for MNIST and 20 times for Fashion-MNIST and CIFAR10. We report the mean accuracy along with the standard deviation. Figure 2 shows the test accuracy of our loss compared to the cross-entropy loss and hinge loss over different training set sizes for MNIST (a), Fashion-MNIST (b) and CIFAR10 (c). Our loss function outperforms the standard methods.

For the implementation of Equation 11 we used $\alpha = -1$, d $= L_\infty$, and approximated d$(x, \mathcal{S}_t) \approx \|x - p^t\|_\infty$, where $p^t$ denotes the projection of $p$ on the level set $\mathcal{S}_t$, $t \in \{-1, 0, 1\}$ (see Section 2.1). The approximation of the term $\Delta(x; \theta)$, where $x = x_j$ is a train example, is therefore $\min\{\|p^0 - p^{-1}\|_\infty, \|p^0 - p^1\|_\infty\}$. See supplementary material for further implementation details.

## 5.2 Robustness to adversarial examples

In this experiment we used our method with the loss in Equation 12 to train robust models on MNIST [18] and CIFAR10 [16] datasets. For MNIST we used ConvNet-4a (312K params) used in [32] and for CIFAR10 we used two architectures: ConvNet-4b (2.5M params) from [30] and ResNet-18 (11.2M params) from [32]. We report results using the loss in Equation 12 with the choice of $\varepsilon_j$ fixed as $\varepsilon_{\text{train}}$ in Table 1, $\lambda_j$ to be 1, 11 for MNIST and CIFAR10 (resp.), and d $= \rho$ as explained in Section 3.2. We evaluated our trained networks on $L_\infty$ bounded attacks with $\varepsilon_{\text{attack}}$ radius using Projected Gradient Descent (PGD) [17, 20] and compared to networks with the same architectures trained using the methods of Madry et al. [20] and TRADES [32]. We found our models to be robust to PGD attacks based on the Xent loss; during the revision of this paper we discovered weakness of our trained models to PGD attack based on the margin loss, i.e., $\min\{F^j\}$, where $F^j = f_j - \max_{i \neq j} f_i$; we attribute this fact to the large gradients created at the level set. Consequently, we added margin loss attacks to our evaluation. The results are summarized in Table 1. Note that although superior in robustness to Xent attacks we are comparable to baseline methods for margin attacks. Furthermore, we believe the relatively low robust accuracy of our model when using the ResNet-18 architecture is due to the fact that we didn't specifically adapt our method to Batch-Norm layers.

In the supplementary material we provide tables summarizing robustness of our trained models (MNIST ConvNet-4a and CIFAR10 ConvNet-4b) to black-box attacks; we log black-box attacks of our and baseline methods [20, 32] in an all-versus-all fashion. In general, we found that all black-box attacks are less effective than the relevant white-box attacks, our method performs better when using standard model black-box attacks, and that all three methods compared are in general similar in their black-box robustness.

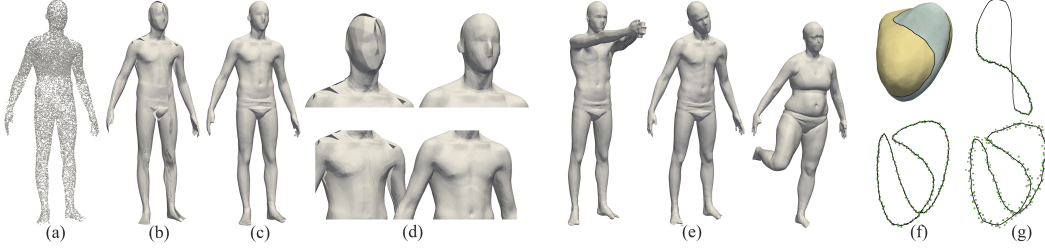

Figure 3: Point cloud reconstruction. Surface: (a) input point cloud; (b) AtlasNet [12] reconstruction; (c) our result; (d) blow-ups; (e) more examples of our reconstruction. Curve: (f) bottom image shows a curve reconstructed (black line) from a point cloud (green points) as an intersection of two scalar level sets (top image); (g) bottom shows curve reconstruction from a point cloud with large noise, where top image demonstrates the graceful completion of an open curve point cloud data.

## 5.3   Surface and curve reconstruction

In this experiment we used our method to reconstruct curves and surfaces in $\mathbb{R}^3$ using only incomplete point cloud data $\mathcal{X} \subset \mathbb{R}^3$, which is an important task in 3D shape acquisition and processing. Each point cloud is processed independently using the loss function described in Equation 13, which encourages the zero level set of the network to pass through the point cloud. For surfaces, it also moves other level sets to be of the correct distance to the point cloud.

For surface reconstruction, we trained on 10 human raw scans from the FAUST dataset [5], where each scan consists of $\sim$ 170K points in $\mathbb{R}^3$. The scans include partial connectivity information which we do not use. After convergence, we reconstruct the mesh using the marching cubes algorithm [19] sampled at a

|  | Chamfer L1 | Chamfer L2 |
|---|---|---|
| AtlasNet-1 sphere | $23.56 \pm 2.91$ | $17.69 \pm 2.45$ |
| AtlasNet-1 patch | $18.67 \pm 3.45$ | $13.38 \pm 2.66$ |
| AtlasNet-25 patches | $11.54 \pm 0.53$ | $7.89 \pm 0.42$ |
| Ours | $\mathbf{10.71} \pm 0.63$ | $\mathbf{7.32} \pm 0.46$ |

Table 2: Surface reconstruction results.

resolution of $[100]^3$. Table 2 compares our method with the recent method of [12] which also works directly with point clouds. Evaluation is done using the Chamfer distance [11] computed between 30K uniformly sampled points from our and [12] reconstructed surfaces and the ground truth registrations provided by the dataset, with both $L_1, L_2$ norms. Numbers in the table are multiplied by $10^3$. We can see that our method outperforms its competitor; Figure 3b-3e show examples of surfaces reconstructed from a point cloud (a batch of 10K points is shown in 3a) using our method (in 3c, 3d-right, 3e), and the method of [12] (in 3b, 3d-left). Importantly, we note that there are recent methods for implicit surface representation using deep neural networks [6, 24, 22]. These methods use signed distance information and/or the occupancy function of the ground truth surfaces and perform regression on these values. Our formulation, in contrast, allows working directly on the more common, raw input of point clouds.

For curve reconstruction, we took a noisy sample of parametric curves in $\mathbb{R}^3$ and used similar network to the surface case, except its output layer consists of two values. We trained the network with the loss Equation 13, where $\mathcal{T} = \{0\}$, using similar settings to the surface case. Figure 3f shows an example of the input point cloud (in green) and the reconstructed curve (in black) (see bottom image), as well as the two hyper-surfaces of the trained network, the intersection of which defines the final reconstructed curve (see top image); 3g shows two more examples: reconstruction of a curve from higher noise samples (see bottom image), and reconstruction of a curve from partial curve data (see top image); note how the network gracefully completes the curve.

## 6   Conclusions

We have introduced a simple and scalable method to incorporate level sets of neural networks into a general family of loss functions. Testing this method on a wide range of learning tasks we found the method particularly easy to use and applicable in different settings. Current limitations and interesting venues for future work include: applying our method with the batch normalization layer (requires generalization from points to batches); investigating control of intermediate layers' level sets; developing sampling conditions to ensure coverage of the neural level sets; and employing additional geometrical regularization to the neural level sets (e.g., penalize curvature).

**Acknowledgments**

This research was supported in part by the European Research Council (ERC Consolidator Grant, "LiftMatch" 771136) and the Israel Science Foundation (Grant No. 1830/17).

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
