[Supplementary Material]

# Controlling Neural Level Sets

**Matan Atzmon, Niv Haim, Lior Yariv, Ofer Israelov, Haggai Maron, Yaron Lipman**
Weizmann Institute of Science
Rehovot, Israel

## 1  Additional Experiments

| Initialization Method | Chamfer | Hausdorf |
|---|---|---|
| Uniform [-0.35,0.35] | 0.011 | 0.141 |
| Normal $\sigma = 0.01$ | 0.006 | 0.017 |
| Normal $\sigma = 0.05$ | 0.01 | 0.132 |

(a) Normal $\sigma = 0.05$     (b) Uniform

Figure A1: Distribution of samples on $d = 2$ neural level set.

### 1.1  Distribution of points on the level set.

Achieving well distributed samples of neural level set is a challenge, especially for high dimensions. In the inset we quantify the quality of distribution in low dimension, $d = 2$, (where ground truth dense sampling of the level set is tractable). The table in Figure A1 logs the Chamfer and Hausdorff distances of the resulting sampling distribution and the level set of a neural network trained with Xent loss in 2 dimensions ((a) and (b)) where projected points (red) are initialized using a uniformly distributed points (gray, (b)) or normally perturbed level set samples (gray, (a)).

Figure A2: Level sets of a network $F$ from the experiment described in Section 5.3 shown along a cross-cut. Note how the iso-levels are equispaced, as encouraged by the loss in Equation 13.

### 1.2  Level sets of reconstruction networks resemble signed distance function

Figure A2 shows iso-levels of one of the networks from the experiment described in Section 5.3. Note how the level sets resemble the level sets of a signed distance function.

# 2 Implementation details

All experiments are conducted on a Tesla V100 Nvidia GPU using PYTORCH framework [6].

| ConvNet-2a | ConvNet-2b | ConvNet-4a | ConvNet-4b | FC-1 | FC-2 |
|---|---|---|---|---|---|
| CONV 16 4x4+2 | CONV 32 5x5+1 | CONV 32 3x3+1 | CONV 32 3x3+1 | FC 512 | FC 509 |
| CONV 32 4x4+2 | MAXPOOL 2x2 | CONV 32 3x3+1 | CONV 32 4x4+2 | FC 512 | FC + SKIP 509 |
| FC 100 | CONV 64 5x5+1 | MAXPOOL 2x2 | CONV 64 3x3+1 | FC 512 | FC + SKIP 509 |
| FC 10 | MAXPOOL 2x2 | CONV 64 3x3+1 | CONV 64 4x4+2 | FC 1 | FC + SKIP 509 |
| | FC 512 | CONV 64 3x3+1 | FC 512 | | FC + SKIP 509 |
| | FC 10 | MAXPOOL 2x2 | FC 512 | | FC + SKIP 509 |
| | | FC 200 | FC 10 | | FC + SKIP 509 |
| | | FC 200 | | | FC 1 |
| | | FC 10 | | | |

Table A1: Our architectures. CONV $kw \times h + s$ corresponds to a convolution layer with $k$ channels, a kernel of size $w \times h$ and stride $s$. FC $n$ correspond to a fully connected layer with $n$ outputs. FC + SKIP indicates a skip connection to the input layer. Each CONV/FC layer is followed by a ReLU activation except for the last fully connected layer.

## 2.1 Parameters of experiments shown in Figure 1

We train a 4-layer MLP $F(x; \theta) : \mathbb{R}^2 \times \mathbb{R}^m \to \mathbb{R}^2$, as in architecture FC-1, for 1000 epochs using the ADAM optimizer [4] with learning rate 0.001. For the geometrical SVM loss we use $\lambda = 0.001$. Training set is composed of 16 points in $\mathbb{R}^2$, all of which lie inside $[0, 0.5]^2$. Batch size is 1. The sample network makes a maximum of 20 iterations for the projection procedure.

## 2.2 Classification generalization

In Table A2 we summarize all hyper-parameters used in the generalization experiments (Section 5.1). For cross-entropy and hinge losses we checked learning rates of 0.001, 0.005, 0.01, 0.02 and chose the ones that performed best. All models were trained using SGD (Nesterov) optimizer with momentum 0.9 and weight decay $10^{-4}$.

| Dataset / Params | MNIST | Fashion-MNIST | CIFAR10 |
|---|---|---|---|
| Architecture | ConvNet-2b | ConvNet-2b | ConvNet-4b |
| Geometric SVM $\lambda$ | $\lambda$ grows linearly from 0.01 to 0.2 over 50 epochs | $\lambda$ grows linearly from 0.01 to 0.2 over 50 epochs | $\lambda$ grows linearly from 0 to 0.01 over 50 epochs |
| # Epochs | 200 | 200 | 100 |
| Batch size | 256 (32 for fraction $\leq 0.3$) | 32 | 32 |
| # Iterations for projection proc. | 20 | 20 | 20 |
| Initial learning rate | 0.02 | 0.02 | 0.01 |
| Learning rate decay | multipled by 0.5 at epochs 50, 100, 120, 140, 160, 180 | multipled by 0.5 at epochs 50, 100, 120 | multipled by 0.5 at epoch 50 |

Table A2: Generalization experiments hyperparameters

## 2.3 Adversarial robustness

We describe the parameters used in the experiments shown in Secion 5.2.

**Training parameters**

We use the networks described in Table A1, labeled ConvNet-4a and ConvNet-4b (following [7]) for the MNIST and CIFAR10 experiments respectively. Additionally, for CIFAR10 we add an experiment with ResNet-18 architecture as in [8]. All networks are trained with batches of size 128. For the projection on the zero levelset procedure, we used the false-position method with a maximum of 40 iterations per batch. The standard models are trained using cross-entropy loss for 200 epochs on MNIST and CIFAR10 respectively (batch-size and learning rates are similar to the above mentioned models). All our models are trained using ADAM optimizer [4].

**Bounded Attack (Table 1)**  We use the *advertorch* library [3]. The attacks parameters are, for MNIST: $\varepsilon_{\text{attack}} = 0.3$, PGD-iterations 40 and 100 and step size 0.01. For CIFAR10: $\varepsilon_{\text{attack}} = 8/255$, PGD-iterations 20 and step size 0.003. All models are evaluated at epoch 200, except for Madry defense with ResNet-18 architecture evaluated at epoch 50.

<table>
<tr><td colspan="5" align="center">(a) Robust Accuracy Xent</td><td colspan="5" align="center">(b) Robust Accuracy Margin</td></tr>
<tr><td>Source<br>Target</td><td>Standard</td><td>Madry</td><td>Trades</td><td>Ours</td><td>Source<br>Target</td><td>Standard</td><td>Madry</td><td>Trades</td><td>Ours</td></tr>
<tr><td>Madry</td><td>98.96%</td><td>96.04%</td><td>97.76%</td><td>99.21%</td><td>Madry</td><td>98.95%</td><td>96.11%</td><td>97.81%</td><td>98.78%</td></tr>
<tr><td>Trades</td><td>98.57%</td><td>97.46%</td><td>96.78%</td><td>98.87%</td><td>Trades</td><td>98.56%</td><td>97.5%</td><td>96.74%</td><td>98.46%</td></tr>
<tr><td>Ours</td><td>99.04%</td><td>97.78%</td><td>97.95%</td><td>99.23%</td><td>Ours</td><td>99.04%</td><td>97.87%</td><td>97.99%</td><td>97.35%</td></tr>
</table>

Table A3: MNIST: Comparison of our method and baseline methods under black-box PGD$^{40}$ attack with $\varepsilon_{\text{attack}} = 0.3$. Rows (target) are the attacked models. All models are trained with ConvNet-4a architecture. Diagonal represents white-box attacks.

<table>
<tr><td colspan="5" align="center">(a) Robust Accuracy Xent</td><td colspan="5" align="center">(b) Robust Accuracy Margin</td></tr>
<tr><td>Source<br>Target</td><td>Standard</td><td>Madry</td><td>Trades</td><td>Ours</td><td>Source<br>Target</td><td>Standard</td><td>Madry</td><td>Trades</td><td>Ours</td></tr>
<tr><td>Madry</td><td>61.5%</td><td>41.53%</td><td>49.76%</td><td>50.97%</td><td>Madry</td><td>61.46%</td><td>39.13%</td><td>39.14%</td><td>51.08%</td></tr>
<tr><td>Trades</td><td>67.84%</td><td>54.72%</td><td>41.89%</td><td>53.11%</td><td>Trades</td><td>67.58%</td><td>53.15%</td><td>38.25%</td><td>53.1%</td></tr>
<tr><td>Ours</td><td>68.43%</td><td>56.71%</td><td>54.47%</td><td>38.45%</td><td>Ours</td><td>68.42%</td><td>55.85%</td><td>54.04%</td><td>38.54%</td></tr>
</table>

Table A4: CIFAR10: Comparison of our method and baseline methods under black-box PGD$^{20}$ attack with $\varepsilon_{\text{attack}} = 0.031$. Rows (target) are the attacked models. All models are trained with ConvNet-4b architecture. Diagonal represents white-box attacks.

## 2.4 Surface Reconstruction

We describe the parameters used for the experiments in Section 5.3. For the Faust benchmark, the network architecture is set to FC-2 (similarly to [2, 5]) and is used both for our model and AtlasNet. The optimization is done using the ADAM optimizer, batch size set to 10 and the initial learning rate is set to 0.001 (decreased by half at epochs 500,1500,3500). Some additional implementation details are: first, we set the parameter $\lambda$ in our reconstruction loss to grow linearly from 1 to 5 over 1000 epochs. Next, to generate samples of $\mathcal{S}_t$ we add Gaussian noise ($\sigma = 0.1$) to the input batch, randomly sample half of the points and use it as initialization for the projection procedure to $\mathcal{S}_0$. The other half is used to sample various level sets $\mathcal{S}_t$ (see Equation 13). The number of iterations for the projection procedure is set to 10.

For the curve reconstruction experiment, the architecture used is FC-1 with the minor difference that the last layer output size is 2. The ground truth is generated by randomly sampling 6 points in space and generating a curve passing through the points, using cubic spline interpolation. We generate the input point cloud by sampling the ground truth curve and adding small Gaussian noise. The sample size is 300 and sample points are chosen using Farthest Point Sampling. We generate samples from

$\mathcal{S}_0$ using the same procedure described above with the minor difference that the entire batch is used as initialization for the projection procedure (other, non-zero level sets are not sampled).

## 3 Proofs

**Lemma 1.** *Let $\ell(x) = Ax + b$, $A \in \mathbb{R}^{l \times d}$, $b \in \mathbb{R}^l$, $\ell < d$, and $A$ is of full rank $l$. Then Equation 4 applied to $F(x) = \ell(x)$ is an orthogonal projection on the zero level-set of $\ell$, namely, on $\{x \mid \ell(x) = 0\}$.*

*Proof.* Let $p \in \mathbb{R}^d$ be the starting point. A single generalized Newton iteration (Equation 4) is

$$p^{\text{next}} = p - A^\dagger(Ap + b). \tag{1}$$

First, $p^{\text{next}}$ is indeed on the level set because: $\ell(p^{\text{next}}) = A(p - A^\dagger(Ap + b)) + b = 0$, where we used the fact that $AA^\dagger A = A$, and $AA^\dagger = I$ (since $\text{rank}(A) = l$). Furthermore, from Equation 1 we read that $p^{\text{next}} - p \in \text{Im} A^\dagger$ and therefore $p^{\text{next}} - p \in \text{Im} A^T$. This implies that $p^{\text{next}} - p \perp \text{Ker} A$. But $\text{Ker} A$ is the tangent space of the level set $\{x \mid \ell(x) = 0\}$, so $p^{\text{next}}$ is the orthogonal projection of $p$ on the zero level set of $\ell$. □

**Lemma 2.** *The columns of the solution in Equation 8, namely $D_\theta p(\theta_0)$, are in the orthogonal space to the level set $\mathcal{S}(\theta_0)$ at $p_0$.*

*Proof.* $D_\theta p \in \mathbb{R}^{d \times m}$ describes the speed of $p$ w.r.t. each of the parameters in $\theta$. If we assume $A := D_x F(p; \theta_0)$ is of full rank $l$, which is the generic case, then the Moore-Penrose inverse has the form $A^\dagger = A^T (AA^T)^{-1}$. This indicates that the columns of $D_\theta p(\theta_0) = -A^\dagger D_\theta F(p; \theta_0) \in \mathbb{R}^{d \times m}$ belong to $\text{Im} A^T$, which in turn implies that they are orthogonal to $\text{Ker} A$, which is the tangent space of the level set at the point $p_0$ □

**Theorem 1.** *Any watertight, not necessarily bounded, piecewise linear hypersurface $\mathcal{M} \subset \mathbb{R}^d$ can be exactly represented as the neural level set $\mathcal{S}$ of a multilayer perceptron with ReLU activations, $F : \mathbb{R}^d \to \mathbb{R}$.*

*Proof.* Let $h_i(x) = a_i^T x + b_i = 0$, $i \in [k]$ denote the planes supporting the faces of $\mathcal{M}$ where $a_i$ are chosen to be the outward normals to $\mathcal{M}$. Since $\mathcal{M}$ is watertight, it is the boundary of a $d$-dimensional polytope $P$.

For each $\lambda \in \{-1, 0, 1\}^k$, let $P_\lambda = \cap_{i \in [k]} \{x \mid \lambda_i h_i(x) \geq 0\}$. Simply put, $P_\lambda$ is a polytope that is the intersection of closed half-spaces defined by the some of the hyperplanes $h_i$. Out of all the possible $P_\lambda$'s, we're only interested in those that are contained in $P$, so we define $\Lambda = \{\lambda \mid P_\lambda \subseteq P\}$. Now, we wish to show that every point in the interior of the large polytope necessarily also lies in the interior of some small polytope in our collection, i.e that $\cup_{\lambda \in \Lambda} \mathring{P}_\lambda = \mathring{P}$. So let $x \in \mathring{P}$. There are two cases:

Case 1: $h_i(x) \neq 0 \, \forall i \in [k]$. That is, $x$ does not lie exactly on a hyper-plane. We can take the following polytope $P_\lambda$ which contains $x$ in its interior: $\lambda_i = \text{sign}(h_i(x))$. We note that $\lambda \in \{-1, 1\}^k$, and we call such a polytope *minimal*. We argue that the interior of a minimal polytope is either completely inside $P$ or completely outside it. This is true because otherwise the minimal polytope will contain two points that are on two different sides of some hyper-plane, which is inconsistent with $\lambda \in \{-1, 1\}^k$. In our case, we know that $P_\lambda$ and $P$ both contain $x$ in their interior, so necessarily $P_\lambda \subseteq P$, which means that $\lambda \in \Lambda$.

Case 2: $\exists \{i_1, ..., i_l\} \subseteq [k]$ s.t. $h_i(x) = 0 \, \forall i \in \{i_1, ..., i_l\}$, and $h_i \neq 0 \, \forall i \in [k] \setminus \{i_1, ..., i_l\}$. In this case there is no minimal polytope that contains $x$ in its interior, so let us consider all of the minimal polytopes which contain $x$ on their boundary. Let $P_\mu$ be such a minimal polytope. As previously stated, the interior of $P_\mu$ is either completely inside $P$ or completely outside it, but since $x$ is both on the boundary of $P_\mu$ and in the interior of $P$ then necessarily $P_\mu$ is completely inside $P$, i.e, $P_\mu \subset P$. We are interested in the union of all such minimal polytopes. Note that for such a minimal polytope

$P_\mu$, necessarily $\mu_i = sign(h_i(x))\ \forall i \in [k] \setminus \{i_1, ..., i_l\}$. For $i \in \{i_1, ..., i_l\}$, $\mu_i$ may receive any value in $\{1, -1\}$. Thus, the union of all such minimal polytopes is $P_\lambda$ where:

$$\lambda_i = \begin{cases} 0 & , i \in \{i_1, ..., i_l\} \\ \text{sign}(h_i(x)) & , otherwise \end{cases}$$

which clearly contains $x$ in its interior and is itself contained in $P$ (because it is the union of minimal polytopes which are contained in $P$), i.e $\lambda \in \Lambda$.

We are now ready to define a function which will receive positive values on the interior of $P$, negative values outside of $P$, and will have $\mathcal{M}$ as its levelset:

$$f(x) = \max_{\lambda \in \Lambda} \min_{i \in [k]} \lambda_i h_i(x)$$

$f$ is a piecewise linear function and can, therefore, according to Theorem 2.1 in [1], be encoded as an MLP with ReLU activations. The idea is to build $\max$ operators using linear layers and ReLU via $\max\{a, b\} = \frac{\sigma(a-b)}{2} + \frac{\sigma(b-a)}{2} + \frac{a+b}{2}$, where $\sigma(x) = \max(0, s)$ is the ReLU activation. Using this binary $\max$, one can recursively build the $\max$ of a vector. $\min$ is treated similarly.

$\square$