[Reviews · NeurIPS 2019]

Reviewer 1



This paper proposes to control the level-sets of neural networks. To do so, the paper first proposes ways to find and manipulate the level-sets. The approach can improve both the generalization and the robustness against adversarial examples, by augmenting one more layer to the model. This is a quite interesting idea on using the level-sets. The results are quite solid, in particular on defending the adversarial examples. I have a few detailed comments: 1. How well does the model converge? Is it guaranteed to find level-sets through optimizing (3)? 2. What is the practical speed of training the network due to that we have to get the level-sets per iteration? 3. Could you provide ImageNet results on the adversarial examples part? Would like to adjust my scores if these questions can be resolved during the rebuttal period.

Reviewer 2



This paper addresses the important task of controlling the level sets that comprise the decision boundaries of a neural network. I think the proposed method is quite reasonable, well-described, and convincingly demonstrated to be quite useful across a number of tasks. Re level set sampling: - Doesn't the ReLU activation imply that D_x F(p; theta) is often =0 at many points p? How do you get around this issue when attempting to optimize p toward S(theta)? It seems this optimization might often get stuck in regions where D_x F(p;theta) = 0 yet p lies far away from S(theta). - The choice of sampling distribution for the initial points p_i should be moved from the Appendix to the main text as this seems critical. Furthermore, the particular choice used by the authors should be a bit better motivated in the text as it's not clear to me. - It seems one can sample from level 0 set by instead just optimizing: min_x ||F(x;theta)|| via gradient descent in x. Did the authors try this procedure? It would be good to comment why the authors proposed method is superior (or an actual example where the authors proposed method is truly superior would be even better!) - In general, since the proposed S(theta) sampling procedure does not come with theoretical guarantees, I recommend the authors empirically evaluate their sampling procedure by plotting ||F(p^*, theta)|| where p^* is the solution found after 10-20 Newton iterations. It seems important to verify the samples are actually coming from near the level 0 set, which is not done in the paper (otherwise the overall approach might be working for different mysterious reasons). It would also be nice to visually show some of the samples from S(theta) when the data are images. - In equations (1)-(2): the authors should write f_j(x; theta), f_i(x; theta), and F(x; theta) for clarity. Also, equation (3) seems redundant given (2). Overall, I recommend the authors initially define everything in terms of level C sets instead of level 0 sets, as 0 doesn't really seem to have special significance and might confuse readers. - In Figure 1: I disagree with authors' statement that panel (c) with L2 margin depicts a decision boundary that better explains the training examples compared to (a). First of all "better explains" does not seem to be the appropriate terminology to use here, perhaps "better models" is more appropriate? Secondly, (c) shows an arbitrary red triangle on the lefthand side, in a region that only contains blue training examples, which seems to be a weird artifact of empirical risk minimization; furthermore, the depicted decision boundary is very jagged-looking compared with (a) which seems brittle. Thus, I would say only (b),(d) look better than (a) to me. - The sample network G should be described in a bit greater detail, such as from what space to what space G maps, how it is implemented via a simple fixed linear layer to F(x; theta), etc. - Overall, it would be nice if the authors could provide some intuition on why they believe their proposed methodology works better than other strategies for large-margin deep learning. Update after reading author response: - I am pleased to see the authors practicing thorough science and toning down their robustness claims in the face of these additional experiments. I am not personally so concerned about the potentially-diminished adversarial robustness results, as level-set controlĀ of a model certainly has other important applications. - Re optimizing min_x ||F(x,theta)|| via alternative optimization methods like gradient descent or LBFGS: I encourage the authors to provide wall-clock time comparison to help readers understand the practical gain of the proposed methodology. In nonconvex settings, it is certainly not always the case that Newton steps are practically superior. - I like the idea of comparing with Elsayed et al (2018) and think the conceptual sentence from your rebuttal that establishes the connection between their methodology and yours should be also included in the final paper.

Reviewer 3



The paper has a few conceptual shortcomings: There is no guarantee that the iteration in Eqn 4 would successfully sample a point on the level set. A good distribution of points on the level set should also account for local geometry, e.g., curvature, which is not addressed in the proposed method. A sparse set of samples may not provide adequate control over the behavior of the entire level set. Some of the theoretical results, e.g., Lemma 1 and Theorem 1 do not strike as particularly surprising. Experimentally, the geometric SVM loss appears to lose its edge as soon as a little bit of data is collected (in small sample regimes one would need to resort to stronger priors or go e.g., semi-supervised route in any case). Still, the empirical results in the robust learning setting and surface reconstruction seem to be promising. Line 144: S(theta_0) should be S(theta)?

[Author Response · NeurIPS 2019]

We thank the reviewers for their comments and suggestions. We will incorporate the given technical suggestions in the final version of the paper. Below we address the main concerns raised in the reviews.

**Adversarial Robustness application.** To avoid the widespread phenomenon of breaking allegedly robust training methods shortly after their publication, we decided to further stress test our method with an assortment of adversarial attacks, and found some vulnerabilities of our trained models to direct decision boundary (ddb) attacks and some black-box attacks. Consequently, we restricted some of the Newton projections to be in the direction of PGD-found examples. We performed extensive all vs. all PGD black-box attacks using both ddb and cross-entropy (Xent). Results for MNIST are shown in Table; we log test accuracy where each column represents different attack, diagonal entries are white-box and off-diagonal are black box attacks; right column shows worst-case for each training method (rows). Note that this procedure came with the cost of a net decrease of our performance for white-box attacks, however, we still remain SoTA or comparable. We will update the paper accordingly (including CIFAR10 results) and tone down some of the robustness claims.

| ddb/Xent | Xent | Madry | Trades | Our | Minimum |
|---|---|---|---|---|---|
| Madry | 98.6/98.6 | 96.1/96.3 | 97.9/97.9 | 98.7/99.1 | 96.1/96.3 |
| Trades | 98.6/98.6 | 98.5/98.5 | 96.9/96.7 | 98.8/99.2 | 96.9/96.7 |
| Our | 98.6/98.5 | 98.2/98.3 | 97.8/97.7 | 96.7/98.6 | 96.7/97.7 |

**(R1) "How well does the model converge? Is it guaranteed to find level sets through optimizing (3)?";**
**(R3) "There is no guarantee that the iteration in Eq. 4 would successfully sample a point on the level set."**
Newton's method is not guaranteed to find zeros of non-linear functions. Although ReLU networks do not satisfy the conditions required for Newton's quadratic convergence it still works well in practice. Empirically, we applied ten Newton iterations and converged to the zero level set between 80-90% of the times (manifold reconstruction and early robust trainings) to 20-30% (end of robust training). Note that even when the Newton projection fails we can use it with non zero $c$ (see Eq. (9)), which is useful for manifold reconstruction.

**(R1) "What is the practical speed of training the network due to that we have to get the level sets per iteration?"**
When comparing training times with level set sampling phase and without we get $\times 2$ the time for manifold reconstruction and $\times 8$ for adversarial training.

**(R2) "Doesn't the ReLU activation imply that $D_x F(p; \theta)$ is often $= 0$ at many points p?"** The last layer is not followed by a ReLU activation, so for $D_x F(p; \theta)$ to be 0 you need all of the neurons from the previous fully-connected layer to be on the zero-region of their respective ReLU activations. Theoretically, if all weights are i.i.d. then chances this happens is $0.5$ to the power of the number of neurons in previous to last layer. Empirically, this doesn't happen.

**(R2) "It seems one can sample from level 0 set by instead just optimizing: $\min_x \|F(x; \theta)\|$ via gradient descent in x. Did the authors try this procedure?"** We have tried the suggested gradient descent (GD) procedure and found it required two orders of magnitude more iterations than Newton projection to converge. Intuitively, the reason Newton is much faster than GD for root finding is that GD linearizes the function at a point and takes a small step toward the zero set, while Newton linearizes the function and goes all the way to the root of the linear function as the next step.

**(R3) "A good distribution of points on the level set should also account for local geometry, e.g., curvature, which is not addressed in the proposed method.".** This is indeed a good point (and a true challenge). From a practical point of view we quantify the quality of distribution in low dimension (where ground truth dense sampling of the level set is tractable). The table logs the Chamfer and Hausdorff distances of the resulting sampling distribution and the level set of a neural network trained with Xent loss in 2 dimensions where projected

| Initialization Method | Chamfer | Hausdorf |
|---|---|---|
| Uniform [-0.35,0.35] | 0.011 | 0.141 |
| Normal $\sigma = 0.01$ | 0.006 | 0.017 |
| Normal $\sigma = 0.05$ | 0.01 | 0.132 |

(a) Normal $\sigma = 0.05$  (b) Uniform

points (red) are initialized using a uniformly distributed points (gray, right) or normally perturbed level set samples (gray, left).

**(R3) "A sparse set of samples may not provide adequate control over the behavior of the entire level set."** Indeed in high dimensions (i.e., not for surface and curve modeling) it would be impossible to densely cover the entire level set with projections since its volume is too large. However, our approach does move the entire level set in the desired manner due to the effect of generalization that is manifested when optimizing a neural network with SGD. This is supported empirically, e.g., the inset shows the histograms of distances of MNIST *test* samples to their projection on the zero-level set of model trained by our method (orange) and a baseline (blue). Note that distances evaluation on the test set means sampling the level set at unseen points.

**(R2) Conceptual discussion (and empirical comparison) on why the proposed approach should work better than other strategies for large-margin deep.** We will add a comparison with a popular large-margin deep model, namely level set linearization methods (e.g., Elsayed et al. [2018]). Conceptually, for $\| \cdot \|_2$, this method is equivalent to working in our framework with a *single* Newton iteration providing only a crude approximation to the neural level set.

[Meta-Review · NeurIPS 2019]

The paper proposes a method to control the level set of the decision surface of neural networks. The reviewers found the approach to be novel and convincingly demonstrated to work on tasks such as CIFAR-10. Furthermore, the method is shown to increase robustness to adversarial perturbations. The work successfully tackles an important topic and as such would be of interesting to the NeurIPS community.